# In Vivo Immune Cell Responses and Long-Term Effects of Cold Atmospheric Plasma in the Upper Respiratory Tract

**DOI:** 10.3390/ijms26188852

**Published:** 2025-09-11

**Authors:** Stephanie Arndt, Petra Unger, Lisa Gebhardt, Robert Schober, Mark Berneburg, Sigrid Karrer

**Affiliations:** 1Department of Dermatology, University Medical Center Regensburg, 93053 Regensburg, Germany; petra.unger@ukr.de (P.U.); mark.berneburg@ukr.de (M.B.); sigrid.karrer@ukr.de (S.K.); 2Terraplasma Medical GmbH, 85748 Garching, Germany; lisa.gebhardt@terraplasma-medical.com (L.G.); schober@terraplasma.com (R.S.)

**Keywords:** cold atmospheric plasma (CAP), plasma intensive care (PIC), pressurized air (PA), upper respiratory tract (URT), bronchoalveolar lavage fluid (BALF), immune cells

## Abstract

Cold atmospheric plasma (CAP) devices produce reactive oxygen and reactive nitrogen species, which have antimicrobial and antiviral effects, while also affecting the molecular and cellular processes in eukaryotic cells. This study investigates the effects of CAP treatment on immune responses and long-term organism health in the upper respiratory tract (URT). Using a surface-microdischarge-based plasma intensive care (PIC) device from terraplasma medical GmbH, *129Sv/Ev* wildtype mice were exposed to short (single 10 min session), long (five 10 min sessions), and recovery-phase treatments (five 10 min sessions; 7 days of recovery). Bronchoalveolar lavage fluid was examined by cytospin, fluorescence-activated cell sorting, and mRNA expression analysis. Lung tissue was analyzed for morphological changes (H&E), DNA damage (γH2AX), apoptosis (TUNEL), immune cell marker alterations (CD45, Ly-6G, CD68, CD3, MCC), and fibrosis (NE). Results showed that PIC treatment increased the number of apoptotic cells and activated immune markers, such as IFN-γ, IL-6, and TNF-α, in the lungs, especially after multiple treatments. These effects largely reversed after a 7-day regeneration period. Importantly, no DNA damage or morphological lung alterations were observed across groups. The findings suggest that PIC treatment in the URT induces transient immune activation without causing tissue damage, but caution is advised for patients with cytokine release syndrome or macrophage activation syndrome due to potential cytokine surges.

## 1. Introduction

Cold atmospheric plasma (CAP) is a partially ionized gas that exhibits potent bactericidal and virucidal properties, making it a promising tool for infection control. Its application in the upper respiratory tract (URT) is particularly intriguing, given its capacity to directly target pathogens within the complex and frequently inaccessible anatomical structures. Unlike traditional liquid antimicrobial agents, CAP can be delivered as a gaseous treatment, allowing it to access areas such as the subglottis, nasal passages, and other body regions that are difficult to reach. This characteristic could enhance the effectiveness of infection prevention strategies, especially in mechanically ventilated patients who are at increased risk of respiratory infections and superinfections.

The potential benefits of CAP in the URT include reducing microbial colonization, preventing biofilm formation on mucosal surfaces, and decreasing the overall microbial load without relying on antibiotics that may contribute to resistance. Additionally, the capability of CAP to generate reactive species can stimulate local immune responses. Its non-invasive nature makes it suitable for repeated applications, which could be advantageous in clinical settings where ongoing infection control is necessary.

A promising area of research lies in the potential of CAP as a novel therapeutic or preventive strategy for hospital-acquired infections, in particular ventilator-associated pneumonia (VAP), a serious and often life-threatening condition prevalent in intensive care units. VAP is defined as pneumonia that develops in patients who have been mechanically ventilated for at least 48 h [1,2,3,4]. It is associated with high mortality rates and an increased reliance on antibiotics, exacerbating the growing problem of antibiotic resistance [2,3,4]. In addition, the financial burden of VAP is significant, resulting in increased hospital costs due to longer hospitalization, additional treatments, and the intensive care required to manage such infections [2,3,5,6].

Previous studies have demonstrated the safe use of CAP in various eukaryotic cells in both in vitro and in vivo experiments [7,8,9], as well as its strong antibacterial efficacy [10,11]. However, its application in sensitive areas, such as the respiratory tract, still requires thorough investigation to avoid potential side effects, particularly excessive immune responses or cytotoxicity caused by reactive oxygen and nitrogen species (RONS) production [12].

Terraplasma medical GmbH (Garching, Germany), which recently developed a method of applying CAP homogeneously through a tube system, has made a key advancement in the medical application of CAP. Plasma is generated by the designated plasma intensive care (PIC) device and is transported via pressurized air (PA) to the treated object. The PIC device allows CAP to reach areas directly within the respiratory tract that are typically inaccessible with conventional therapies.

Using the PIC device and a three-dimensional (3D) model of the URT, Karrer et al. (2024) [13] showed in an in vitro safety study on the treatment of oral keratinocytes, human bronchial/tracheal epithelial cells, and human lung fibroblasts that CAP does not significantly affect cell viability, morphology, apoptosis, DNA damage, or migration. However, immunomodulatory effects, such as the increased release of pro-inflammatory cytokines (MCP-1, IL-6, IL-8, TNF-α), were observed [13]. It is critical to rule out the possibility of an immune overreaction in patients developing cytokine release syndrome (CRS) or macrophage activation syndrome (MAS), as such a response could have life-threatening consequences.

To address these concerns, the study by Reichold et al. (2024) [14] aimed to provide a more comprehensive understanding of how CAP influences immune cells in the context of respiratory infections. The effects were analyzed in vitro in a 3D URT model, focusing on the impact of CAP on the defense mechanisms of polymorphonuclear neutrophils (PMNs) including migration, intracellular ROS production, NETosis, surface antigen expression, and cell death [14]. Their findings indicate that treatment with the PIC device has an impact on the functionality and activity of isolated PMNs.

PIC treatment achieved excellent direct antimicrobial efficacy (99.9%), but it impaired PMN migration by reducing Track Length (TL) and Track Straightness (TS). This suggests compromised PMN motility and movement toward pathogens, potentially diminishing sustained pathogen clearance [14]. The trade-off between rapid bacterial eradication and preservation of longer-term immune function highlights the need for careful consideration of CAP device selection in clinical settings.

Despite high extracellular reactive oxygen and nitrogen species (RONS) after PIC treatment, CAP stimulation did not induce significant intracellular ROS or NETosis in PMNs, indicating CAP exposure did not activate PMN defenses in vitro. On the contrary, receptor-mediated ROS responses to the chemoattractant Formyl-Methionyl-Leucyl-Phenylalanine (fMLP) and TNF-α were reduced post PIC treatment [14].

However, the observed inhibition of receptor-mediated ROS production and the previously mentioned impairments in PMN migration after PIC treatment suggest that the effect of PIC may weaken the antimicrobial defense mechanisms of PMNs, potentially counterbalancing its strong antibacterial efficacy.

For URT clinical use, optimizing plasma parameters to maximize safety and minimize immune overreaction is crucial. Further preclinical studies are needed to refine CAP applications for treating respiratory infections like VAP.

The present study aims to evaluate CAP effects on lung tissue and its effect on immune responses in vivo. Understanding these factors will be crucial for translating CAP technology into routine clinical practice for managing respiratory infections and improving patient outcomes in intensive care units.

## 2. Results

To evaluate the efficacy and safety of CAP treatment in the URT, *129Sv/Ev* wildtype mice were exposed to pressurized air (PA) as control or to CAP, using the plasma intensive care (PIC) treatment setup, where plasma components were directed into a treatment chamber via an airflow (pressurized air; PA) as described in Figure 1 and Section 4.3.

Treatment was performed in a single treatment chamber according to the treatment protocol summarized in Table 1 and Section 4.1. Mice were exposed to short (single 10 min session; group 1B and 1B^1^), long (five 10 min sessions; group 2B and 2B^1^), and recovery-phase treatments (five 10 min sessions; 7 days of regeneration; group 3B and 3B^1^). Control groups (group 1A–3A and 1A^1^–3A^1^) were treated accordingly with PA.

Animals from groups 1A–3A and 1B–3B were processed for BALF analysis as shown in Figure 2 and Section 4.4.1, whereas mice from groups 1A^1^–3A^1^ and 1B^1^–3B^1^ were used for histological lung examinations.

### 2.1. Evaluation of Bronchoalveolar Lavage Fluid (BALF) After PIC Treatment

#### 2.1.1. Cell Cluster Formation and Erythrocyte Adhesion After PIC Treatment

Since the number of isolated cells from the BALF extraction was limited, cytospin preparations were made for both short-term (group 1A, 1B) and long-term (group 2A, 2B) experiments to facilitate microscopic analysis. Results indicated an increase in cell cluster formation and erythrocyte adhesion after PIC treatment (Figure 3a). To achieve more precise immunophenotyping, the long-term experiment, which included a regeneration phase (group 3A, 3B), did not rely solely on cytospin diagnostics. Instead, a targeted immunocytochemical analysis using specific antibodies against various immune cell markers, such as CD68 (macrophages), CD3 (T lymphocytes), CD45 (common leukocytes), and Ly-6G (granulocytes), was performed. This approach enabled a detailed assessment of the cellular composition and provided insights into the relative proportions of different immune cell populations after PA and PIC treatments after a regeneration time of 7 days. The results (Figure 3b) showed no significant differences between PA- and PIC-treated BALF cells, suggesting that any potential changes to the immune cell population that occurred during the 7-day regeneration period had fully normalized.

#### 2.1.2. Apoptosis Detection in BALF Cells After PIC Treatment

Using FACS PI/Annexin V analysis, apoptosis, necrosis, and the number of living cells were determined after PA (group 1A–3A) and PIC (group 1B–3B) treatment. Necrosis was virtually undetectable in all treatment groups (Figure 4a–c). The total amount of apoptotic cells (early and late apoptosis) was increased after a single 10 min PIC treatment (group 1B; Figure 4a) and was further induced after five PIC treatments of 10 min each (group 2B; Figure 4b). A regeneration period of 7 days after five CAP treatments of 10 min each resulted in a marked decrease in apoptotic cells (group 3B; Figure 4c). However, it should be noted that the results are not statistically significant and therefore only show a trend. While the data suggest a possible effect, the evidence is not strong enough to draw firm conclusions. In addition, each plot next to the graph illustrates a representative outcome from one mouse per group.

Interestingly, the basal levels of apoptotic cells in the long-term plus regeneration group 3A were higher than in the short-term experiments (group 1A) and the long-term experiment (group 2A). We speculated that differences in experimental conditions (higher room temperature during BALF isolation) might have influenced the apoptotic rates during this experiment.

#### 2.1.3. Cytokine Induction in BALF Cells After PIC Treatment

Interferon-gamma (IFN-γ) expression in lung cells is a key component of the immune response within the respiratory system [15]. The expression of IFN-γ in lung cells plays a crucial role in orchestrating immune defense against pathogens, modulating inflammation, and influencing tissue repair. Particularly multiple PIC treatments (group 2B; Figure 5), induced IFN-γ, which remained elevated after a regeneration period of 7 days (group 3B; Figure 5). However, these results were not statistically significant due to the large variation between individual animals.

### 2.2. Evaluation of Lung Histology After PIC Treatment

#### 2.2.1. Lung Morphology Was Not Changed After PIC Treatment

Hematoxylin and eosin (H&E) staining is a widely used histological technique for the morphological examination of tissue samples. Visualization of H&E staining of lung tissue after PIC treatment (group 1B^1^, 2B^1^, 3B^1^) showed no structural changes and no signs of inflammation or tissue damage in comparison to the control groups (group 1A^1^, 2A^1^, 3A^1^) (Figure 6). However, concrete statements on tissue damage and changes in immune cell status cannot be adequately assessed from the H&E section; consequently, in Section 2.2.2, Section 2.2.3 and Section 2.2.4, targeted analyses addressing these aspects were performed.

#### 2.2.2. DNA Damage in Lung Tissue Was Not Significantly Altered After PIC Treatment

The gamma H2AX (γH2AX) marker is a widely used marker of DNA double-strand breaks (DSBs) in cells, including those in lung tissue [16]. However, in histological sections of lung tissue after PIC treatment (group 1B^1^, 2B^1^, 3B^1^), γH2AX immunostaining was not significantly (ns) altered in comparison to the PA control treatment groups (group 1A^1^, 2A^1^, 3A^1^) (Figure 7).

#### 2.2.3. Cell Apoptosis in the Lungs Was Induced After PIC Treatment

The Terminal deoxynucleotidyl transferase dUTP Nick End Labeling (TUNEL) assay is a technique used to detect DNA fragmentation resulting from apoptosis. TUNEL staining of lung tissue sections revealed a significant increase in apoptotic cells after PIC treatment (group 1A^1^ vs. 1B^1^; 2A^1^ vs. 2B^1^; 3A^1^ vs. 3B^1^) (Figure 8a,b). The increase in positive cells after PIC treatment seems to be dose-dependent (group 1B^1^ vs. 2B^1^) (Figure 8a,b); however, the number of positive cells was reduced again after a regeneration phase of 7 days (group 2B^1^ vs. group 3B^1^). These results suggest that while apoptosis is induced in a dose-dependent manner after both short-term and long-term treatment, the overall level decreases again after a regeneration time, indicating that PIC treatment has reversible cytotoxic effects on lung tissue.

#### 2.2.4. Immune Cell Alterations Were Observed After PIC Treatment

##### The Common Leukocyte Antigen CD45 Was Induced After PIC Treatment

CD45, also known as leukocyte common antigen, is a transmembrane protein expressed on all nucleated hematopoietic cells [17]. In lung tissue, CD45 serves as a key marker for identifying and studying various immune cell populations involved in respiratory immune responses and inflammation. After PIC treatment (group 1B^1^, 2B^1^, 3B^1^), CD45 was significantly increased (Figure 9a,b). This increase was already detectable after a single PIC treatment (group 1B^1^) and appeared to remain elevated even after a 7-day regeneration period (group 3B^1^). Elevated CD45 levels may reflect an ongoing immune response or inflammation triggered by the PIC treatment. Interestingly, the elevation of CD45 could not be detected in BALF cells after a 7-day regeneration period (Figure 3b).

##### The Granulocyte Marker Ly-6G Was Not Detectable in Lung Tissues

Ly-6G is a surface marker primarily expressed on neutrophils [18]. In healthy lung tissue, Ly-6G-positive neutrophils are typically present at low levels. However, during inflammatory responses such as pneumonia or acute lung injury, their numbers can significantly increase. Interestingly, we did not detect any Ly-6G-positive cells in the tissue sections of the PA controls (groups 1A^1^, 2A^1^, 3A^1^) or in the lung tissue of the PIC-treated mice (groups 1B^1^, 2B^1^, 3B^1^) (Figure 10).

##### Neutrophil Elastase (NE) Was Not Detectable After PIC Treatment

NE, an enzyme primarily produced by neutrophils, is an indicator of neutrophilic inflammation and tissue destruction that could potentially set the stage for fibrotic remodeling over time [19]. We were unable to detect any signs of increased NE after PIC treatment (groups 1B^1^, 2B^1^, 3B^1^) (Figure 11), which is consistent with the observations that neutrophils could also not be detected (Figure 10).

##### The Macrophage Marker CD68 Was Not Significantly Altered After PIC Treatment

In the lungs, CD68 is an important marker used to detect alveolar macrophages, which play a central role in immune defense by clearing debris, particles, and pathogens [20]. An increased number or activation of CD68-positive macrophages can indicate inflammation or tissue remodeling, for instance in conditions such as pneumonia, interstitial lung diseases, or chronic obstructive pulmonary disease (COPD). Overall, we observed only a very small number of CD68-positive cells in the lung tissue sections, and their numbers had not changed significantly after PIC treatment (Figure 12a,b).

##### The T Lymphocyte Marker CD3 Is Not Significantly Changed After PIC Treatments

The presence and distribution of CD3-positive cells can provide a close insight into immune responses or inflammation within the lungs [21]. Typically, an increase in CD3-positive T cells indicates an active immune response, while low levels suggest minimal T cell infiltration. The tissue sections of the PA controls (groups 1A^1^, 2A^1^, 3A^1^) as well as the lung tissue of the PIC-treated mice (groups 1B^1^, 2B^1^, 3B^1^) showed only a very low level of CD3-positive cells (max 0–5 per field of view) and no significant changes were detectable (Figure 13a,b). However, it should be noted here that the overall low numbers of positive cells in the selected images may not fully represent all slices for a treatment or control group. Despite this, there is a trend toward a slight increase in CD3 after PIC treatment. PIC treatment (groups 1B^1^, 2B^1^, 3B^1^) likely induces at most a mild inflammatory response when considering the full dataset. The higher apparent inflammation in control group 3A^1^ is due to greater sample-to-sample variability rather than a consistently higher CD3 count. However, quantitative analysis across all sections showed no statistically significant difference in inflammatory cell density, suggesting that T lymphocyte populations are not significantly affected by PIC treatment.

##### Mast Cell Chymase (MCC) Was Induced After PIC Treatment

In lung tissue, MCC is used as a specific marker to identify and study connective tissue-type mast cells involved in various pulmonary pathologies [22,23]. In the present study, MCC induction was primarily observed after multiple PIC treatments (group 2B^1^) (Figure 14a,b) and appeared to be still slightly elevated after a regeneration period of 7 days (group 3B^1^) (Figure 14a,b).

#### 2.2.5. Cytokine Expression Was Changed After PIC Treatment

##### Interleukin-6 (IL-6) Gene Expression Was Induced After PIC Treatment

IL-6 is a pro-inflammatory cytokine that plays a key role in inflammatory responses and immune regulation [24]. The expression of *IL-6* in lung tissue can be influenced by various treatments or stimuli, such as infections, inflammatory processes, or therapeutic interventions. PIC treatment appeared to induce *IL-6* gene expression, especially after multiple treatments (group 2B^1^). However, these effects were reversible after regeneration (group 3B^1^) (Figure 15a).

##### Tumor Necrosis Factor-Alpha (TNF-α) Gene Expression Was Induced After PIC Treatment

TNF-α is a pro-inflammatory cytokine that is primarily produced by macrophages, neutrophils, and other immune cells [25]. Its elevation in the lungs suggests activation of inflammatory pathways, which can be involved in responses to infection, injury, or certain treatments. A PIC-dependent induction of gene expression was also observed here (groups 1B^1^, 2B^1^), which decreased again after a regeneration period of 7 days (group 3B^1^) and returned to the baseline level (group 3A^1^) (Figure 15b). We want to emphasize that only a trend toward increased *IL-6* and *TNF-α* gene expression was observed and that this trend did not reach statistical significance.

## 3. Discussion

This study investigates the safety and effects of cold atmospheric plasma (CAP) with a focus on its use in the upper respiratory tract (URT). Previous research has shown that CAP possesses strong antibacterial properties while remaining harmless to eukaryotic cells and preventing resistance in pathogens [9,10]. These characteristics led to its medical approval for skin treatments. Recent research has shifted toward exploring the potential of CAP for treating respiratory infections, but concerns about possible immune overreactions remain, given prior observations of immune responses after CAP exposure [13,14]. To address these concerns, this study used *129Sv/Ev* wildtype mice exposed to different CAP treatment regimens—short, long, and with recovery phases—and assessed their effects on lung tissue and cells from bronchoalveolar lavage fluid (BALF) through molecular and histological analyses. The findings aim to determine whether CAP can be safely applied in the URT without provoking harmful immune responses.

Cytospin preparations and immunocytochemical analysis were performed to examine immune cell responses in BALF after plasma intensive care (PIC) treatment in comparison to pressurized air (PA) (Figure 3). Results showed increased cell clustering and erythrocyte adhesion, indicating that PIC may induce oxidative stress and cellular activation, leading to changes in cell surface properties (e.g., integrins, selectins). Other studies have already demonstrated that CAP can modulate the expression of adhesion molecules such as integrins and selectins. For instance, Kupke et al. (2021) [26] found that CAP treatment increases the expression of integrins (CD11b, CD66b) while decreasing the expression of L-selectin (CD62L) on human granulocytes. Their findings suggest that CAP influences the adhesion capacity of granulocytes and potentially their activation status, which could be relevant for applications in wound healing and immunomodulation [26].

Conversely, Schmidt et al. (2017) observed a general downregulation of multiple integrin proteins in fibroblasts after CAP exposure [27]. This reduction in integrin expression may account for morphological changes seen in fibroblasts post-treatment, such as cell rounding, formation of shorter and smaller filopodia, disruption of microfilaments, increased motility, and surface blebbing [27].

Overall, these studies highlight the cell type-specific effects of CAP on adhesion molecules and cellular morphology, underscoring its complex influence on cellular behavior.

FACS PI/Annexin V analysis revealed that PIC treatment increased apoptosis in BALF cells, especially after multiple treatments, while necrosis remained negligible across all groups (Figure 4). Notably, a 7-day regeneration period reduced apoptotic cell levels, suggesting activation of repair mechanisms or clearance of apoptotic cells. However, it should be noted that all the obtained FACS results are not statistically significant and therefore only show a trend. Interestingly, basal apoptosis was higher in the long-term plus regeneration group, possibly due to differences in experimental conditions such as room temperature. In terminal BALF procedures, instillation of room temperature BALF is considered adequate for most applications. However, the review of the methods and uses of BALF by Poitout-Belissent et al. (2021) suggests that cooling BALF to 4 °C may be beneficial in slowing the onset of apoptosis and keeping the viable cell population intact [28]. Overall, our findings indicate that CAP-induced apoptosis is transient and reversible, supporting its safety for application in the respiratory tract.

Additionally, an increase in IFN-γ expression was observed in BALF-isolated cells after multiple PIC treatments, with elevated levels persisting even after a 7-day recovery period (Figure 5). Although these differences were not statistically significant due to individual variability, they suggest that CAP may stimulate immune responses that could enhance antimicrobial defense. Nonetheless, careful monitoring is essential to balance potential therapeutic benefits with the risk of excessive inflammation.

H&E staining of lung tissue after PIC treatment showed no observable structural alterations, inflammation, or tissue damage compared to the control groups (Figure 6). This finding indicates that PIC treatment does not induce histological damage to lung tissue under the conditions studied.

Further, γH2AX immunostaining in lung tissue showed no significant increase in DNA double-strand breaks after PIC treatment compared to PA controls (Figure 7). This finding indicates that, under the studied conditions, CAP therapy does not induce genotoxicity in lung cells, supporting its safety and potential for medical applications without compromising genetic integrity.

TUNEL assay results showed a dose-dependent increase in apoptosis immediately after PIC treatment, with levels decreasing after a 7-day regeneration period (Figure 8). This finding suggests that PIC induces reversible, transient cytotoxic effects on lung tissue, with apoptosis subsiding over time, indicating a potential for safe therapeutic application.

The observed increase in CD45 expression after PIC treatment (Figure 9) indicates an activation or infiltration of immune cells within the lung tissue, consistent with an ongoing immune response or inflammation. Notably, this elevation was detectable after a single dose and persisted throughout the regeneration period, suggesting sustained immune activity. Interestingly, after a seven-day recovery period, CD45 levels in the BALF (Figure 3b) were no longer elevated, indicating that the PIC-induced immune response lasts longer in the lung parenchyma and does not appear to affect inflammatory processes at the luminal or alveolar levels. Comparing this with bronchial biopsies could be informative, since differences in sensitivity between fluids and tissue, as well as measurement methods, can influence outcomes. Therefore, the absence of change in BALF does not rule out potential effects in bronchial tissue.

Although neutrophil infiltration is expected during inflammation, Ly-6G-positive cells were not detected in either control or PIC-treated lung tissues across all groups (Figure 10). This finding suggests that PIC treatment did not induce neutrophil recruitment or activation in this model, indicating a limited neutrophilic inflammatory response.

Neutrophil elastase (NE) levels also remained undetectable across all PIC-treated groups (Figure 11), aligning with the absence of neutrophil infiltration observed in the tissue sections. This finding indicates once more that PIC treatment did not provoke neutrophilic inflammation or associated tissue damage in this in vivo model. The study by Reichold et al. (2024) [14] investigated the cellular responses of polymorphonuclear neutrophils (PMNs) in a three-dimensional (3D) model of the URT after exposure to CAP. In contrast to the present study, in which neutrophils in the lungs were not affected by CAP, the in vitro study by Reichold et al. showed an impact of CAP on the functionality and activity of PMNs isolated from human blood [14]. However, in their in vitro study, the isolated cells were directly treated with CAP, whereas in the present in vivo study, the animals inhaled CAP, which likely resulted in a more indirect effect on the relevant cells. In vitro studies can reveal important mechanisms; however, their translatability to the whole organism is sometimes limited. Therefore, these preclinical in vivo findings are of essential importance for subsequent analysis and the application in patients.

Furthermore, PIC treatment did not significantly alter the number of CD68-positive alveolar macrophages (Figure 12), indicating no notable macrophage-mediated inflammation or tissue remodeling.

Even though the results of the CD3 assessment suggest a mild inflammatory response after PIC treatment (Figure 13), it should be noted that these results indicate only a trend due to the high variability among individual samples and do not provide statistically significant findings. Therefore, these results should be interpreted with caution.

Interestingly, MCC levels increased primarily after multiple PIC treatments, with a slight elevation persisting after 7 days of regeneration (Figure 14). While short-term MCC rise may be normal, sustained or high levels could indicate ongoing inflammation or tissue damage, warranting further investigation.

Finally, PIC treatment induced *IL-6* and *TNF-α* gene expression in the lung, particularly after multiple treatments (Figure 15), indicating an inflammatory response. These mRNA expression levels decreased after a regeneration period, suggesting that the inflammation is reversible and resolves over time. However, the results only show a trend, as they did not reach statistical significance.

Based on the presented data, PIC treatment does not induce significant structural damage or DNA double-strand breaks in lung tissue, indicating a lack of overt toxicity. Although an immediate increase in apoptosis and immune cell activation occurs after treatment, these effects are transient and diminish after a regeneration period. The minimal impact on T lymphocyte populations further suggests that adaptive immune responses remain largely unaffected.

Overall, PIC treatment appears to elicit a temporary inflammatory response that resolves over time without causing any lasting tissue damage, supporting its potential safety profile for therapeutic applications. Although inflammatory indicators had decreased by the end of the recovery period, they remained detectable. In other words, signs of inflammation did not completely vanish and could still be measured, even after the formal recovery interval. Extending the recovery period in future studies could help clarify whether these residual inflammatory signals eventually normalize or persist over a longer term.

## 4. Materials and Methods

### 4.1. Animals

Female *129Sv/Ev* wild type mice (4–8 weeks old; originally obtained from the Robertson Laboratory; Department of Molecular and Cellular Biology, Harvard University, Cambridge, MA, USA) were maintained under specific pathogen-free and controlled conditions (22 °C, 55% humidity, and 12 h day/night rhythm) and had free access to water and chow. The mice received humane care in compliance with the guidelines outlined in the Guide for the Care and Use of Laboratory Animals. The study was approved by the Research Ethics Committee (approval number: 55.2.2-2532-2-1341-22) from the government of Lower Franconia, Würzburg, Germany. Using G*power 3.1.9.4, the minimal group size for a *t*-test at alpha error 0.05 and power of 0.8 and effect size of 1.9 was calculated to be 6 mice per group. All animals (n = 72) were divided into control groups (n = 6 mice per group; 1A–3A and 1A^1^–3A^1^) and experimental groups (n = 6 mice per group; 1B–3B and 1B^1^–3B^1^). After a week of acclimatization in the group cages (n = 3 animals per cage), the URT treatment with the plasma intensive care (PIC) device or the control treatment with pressurized air (PA) was started using the treatment regimen shown in Table 1 (Section 2) and subsequent evaluation. Bronchoalveolar lavage fluid (BALF) (Section 4.4) was evaluated by cytospin, fluorescence-activated cell sorting (FACS), and mRNA expression analysis. For histological evaluation (Section 4.5), lung tissue slices were used.

### 4.2. Plasma Device

A prototype of the plasma care^®^ device developed by terraplasma GmbH, Garching, Germany, was used for CAP treatment. A recent publication detailed the design, technology, and ozone emission spectrum of this device [26]. This prototype enables frequency adjustments between an oxygen mode (4 kHz) and a nitrogen mode (8 kHz), with a 4 kHz setting specifically used in this study due to its strong antibacterial effect. The device employs a technology known as “thin-film technology”, which is an advanced version of surface micro-discharge technology (SMD) [29]. The application of a high voltage of 3.5 kV generates millimeter-sized micro-discharges within the plasma-source unit. The unit consists of a high-voltage electrode, a dielectric, and a grounded structured electrode [26], which together produce plasma components that can be adjusted based on frequency and voltage. In the present work, the plasma care^®^ device, which was originally designed for wound treatment and delivers plasma components to the wound via diffusion, has been modified for application in the URT. In this modification, the plasma components are transported through an airflow (pressurized air; PA). This setup, referred to as the plasma intensive care system (PIC), is described in more detail in Section 4.3. and was provided to us by the company terraplasma medical GmbH (Garching, Germany).

### 4.3. Treatment of Mice with Cold Atmospheric Plasma or Pressurized Air

The mice were divided into different treatment groups (Table 1) and treated either with the plasma intensive care (PIC) device setup (Figure 1a) or with pressurized air (PA) as a control. In the control group, the complete treatment assembly was used, but the PIC device was turned off. The setup includes a function generator (Figure 1b; purple arrow) to set the frequency (4 kHz), voltage (3.5 kV), period (200 ms), and number of cycles (140). The parameter “period (200 ms)” defines the total duration of one cycle, including both active and inactive phases, while the “number of cycles (140)” specifies the electrical alternating current (AC) cycles within the active phase. These settings ensure intermittent operation of the plasma source, which prevents excessive plasma concentrations during inhalation and allows the 10 min treatment duration to be achieved under safe exposure conditions. The plasma source (Figure 1c; red arrow) is connected via two adapter hoses: one hose linked to the compressed air generator (Figure 1c; blue arrow) and another connected to the mouse treatment chamber (Figure 1d; yellow arrow; custom-made by terraplasma medical GmbH, Garching, Germany). An ozone sensor (Figure 1c; green arrow) is installed between these components to detect the ozone concentration immediately before entering the treatment chamber. The ozone levels fluctuate depending on ambient air humidity but should remain within a range of 5–6 ppm, not falling below or exceeding this value.

For treatment, the mice were individually placed under the 3D-printed treatment chamber. Control animals were treated with pressurized air (PA) only, according to the treatment regimen (Table 1), in groups 1A–3A.

Group 1B received CAP treatment with the PIC device one-time for 10 min. In contrast, group 2B was CAP treated five times per week (Monday–Friday) for 10 min each. After each treatment, the animals were returned to their housing group cages. Group 3B also received CAP treatment five times per week (Monday–Friday); however, no immediate tissue collection or BALF analysis was performed after the final treatment. Instead, these groups underwent a 7-day regeneration period.

To ensure an even distribution of CAP components into the treatment chamber, a flowmeter and supply tube connection were used to set a compressed air flow rate of 0.5 standard liters per minute (slm) using a rotary control on the compressed air generator.

The treatment is not stressful for the animals, and no behavioral abnormalities were observed following PIC or PA treatment.

Since BALF procedure can cause some lung damage, separate experimental and control groups were performed for histological evaluation of lung tissue (groups 1A^1^–3A^1^ and 1B^1^–3B^1^).

### 4.4. Bronchoalveolar Lavage Fluid (BALF) Evaluation

The BALF model is an experimental procedure used to examine the cellular and acellular contents of the lung lumen ex vivo, in order to analyze the effects of exogenous factors, e.g., CAP on immune cells [30]. Immune cells and fluids obtained from BALF can be analyzed using various biological and histological methods.

#### 4.4.1. Bronchoalveolar Lavage Fluid (BALF) Extraction

BALF (Figure 2) was extracted immediately after the last PA (group 1A, 2A) or PIC (group 1B, 2B) treatment or after a regeneration time of 7 days (group 3A, 3B). After weighing, the animals received an overdose of pentobarbital (Narcoren^®^, Merial GmbH, Hallbergmoos, Germany) intraperitoneally (i.p.) for euthanasia, adjusted to their body weight. The dosage for painless euthanasia of small rodents is 200–400 mg/kg body weight (100 mL Narcoren^®^ contains 16.0 g of pentobarbital sodium, i.e., 1.25–2.5 mL/kg body weight). Therefore, with 0.06 mL, a mouse weighing approximately 30 g can be euthanized painlessly.

Immediately after death, the animal was placed on a surface in a dorsal position and fixed with adhesive tape to the feet and with a rubber band through the mouth (Figure 2a). After disinfecting the chest area with 70% isopropanol, a vertical incision approximately 5–7 cm long was made in the skin above the thymus, followed by spreading of the tissue until the esophagus and trachea became visible. The skin and muscle tissue covering the trachea were surgically removed (Figure 2a).

Next, a medical suture (Seraflex 4/0, Serag/Wiessner, Naila, Germany) was inserted under the trachea (Figure 2b), and the trachea was then tied to the tracheal tube. Using a microscope, a horizontal incision was made between two tracheal rings. The tracheal tube was then inserted into this incision and carefully advanced into the airways until resistance was felt. The tube was secured to the trachea with a surgical knot, and lavage was started by injecting 800 μL of Dulbecco’s Phosphate-Buffered Saline (DPBS) (Thermo Fisher Scientific, Waltham, MA, USA) through a syringe inserted into the tracheal tube (Figure 2c). During this process, the mouse’s chest was gently massaged to maximize BALF collection.

The BALF was collected in a 2 mL Eppendorf™ collection tube (Thermo Fisher Scientific, Waltham, MA, USA), and the lavage procedure was repeated two more times. In total, approximately 2 mL of BALF could be obtained per animal. For analysis, BALF was centrifuged for 5 min at about 3000 rpm to separate immune cells from fluids. Cells were than resuspendated with 1 mL DPBS and counted. The immune cells were immediately processed for further analysis (Section 4.4.2, Section 4.4.3 and Section 4.4.4).

However, this lavage procedure causes significant damage to the lung tissue. Therefore, experiments intended for histological examination of the lung after PIC or PA treatment had to be performed separately.

#### 4.4.2. Cytospin Preparation and Imaging

Cytopsins were prepared from PA- (group 1A, 2A) and PIC- (group 1B, 2B) treated animals by centrifugating the cellular fraction of the BALF (approximately 500,000 cells/250 µL DPBS) by 1200 rpm for 10 min onto Superfrost Plus Adhesion Microscope Slides (Epredia, Breda, The Netherlands). The slides were carefully removed from cytocentrifuge and allow to air dry prior to staining. Afterwards, cells were stained with May-Grünwald Giemsa (Sigma-Aldrich, Taufkirchen, Germany) according to standard practices [31]. The cytospin preparations were microscoped at 40× magnification using a transmitted light microscope (Carl Zeiss Vision GmbH, Halbergmoos, Germany) and three images per slide were collected.

#### 4.4.3. Leukocyte Profile Evaluated by Flow Cytometry

1 × 10^6^ cells from the BALF were collected from PA- (group 3A) and PIC- (group 3B) treated animals immediately upon extraction and analyzed by flow cytometry for the differentiation of leukocytes. After centrifugation for 3 min at 1500 rpm at 4 °C, the supernatant was removed. The cells were immediately stained in 5 mL Sarstedt Round Bottom Polystyrene Tubes (Thermo Fisher Scientific, Waltham, MA, USA) with an antibody cocktail containing 5 µL CD68 (E3O7V) rabbit monoclonal antibody (Alexa Fluor^®^ 488 Conjugate; FL1-H), 5 µL CD3 (17A2) rat monoclonal antibody (PE Conjugate; FL2-H), 5 µL CD45R/B220 (RA3-6B2) rat monoclonal antibody (PerCP-Cy5.5^®^ Conjugate; FL3-H), and 5 µL Ly6G (1A8) rat monoclonal antibody (APC Conjugate; FL4-H) each from Cell Signaling Technology, Frankfurt, Germany, and incubated for 15 min at 4 °C in darkness. After washing with DPBS, cells were again resuspended in 200 µL DPBS. Immediately, a minimum of 5000 events were acquired using a FACS Calibur Flow Cytometer (Becton Dickinson, Heidelberg, Germany). Data were analyzed using the FlowJo™ v10 software.

#### 4.4.4. Measurement of Cell Apoptosis and Necrosis by Flow Cytometry

For analysis of live, early apoptotic, late apoptotic, and necrotic cells, 1 × 10^6^ cells from the BALF were investigated by flow cytometry using the FITC Annexin V Apoptosis Detection Kit with PI (BioLegend, Koblenz, Germany) according to the manufacturer’s instructions. Analysis was done with a FACS Calibur Flow Cytometer (Becton Dickinson, Heidelberg, Germany). Live cells were characterized as both PI and Annexin V negative. Annexin V only positive cells were early apoptotic and both PI and Annexin V positive cells were late apoptotic. Cells stained only for PI were necrotic. FACS data were analyzed using the FlowJo™ v10 software. Data are representative of n = 6 animals per group (group 1A–3A and 1B–3B).

### 4.5. Histological Evaluation

Histological examination was conducted on formalin-fixed and paraffin-embedded preparations from group 1A^1^–3A^1^ and 1B^1^–3B^1^ (n = 36). For the analysis of the lung, small tissue samples (~0.5 cm^2^) were taken from each lobe of the right lung: the superior lobe, middle lobe, inferior lobe, and post-caval lobe. As immunohistochemical positive controls, various murine tissue sections from previous studies were used. These specimens were co-stained according to the antibody positivity for the corresponding stains of the lung sections, following the protocols described in Section 4.5.2, Section 4.5.4 and Section 4.5.5. The immunohistochemical positive controls for the assays are summarized here (Figure 16).

#### 4.5.1. Determination of Morphological Changes (H&E)

Tissue sections of the lungs measuring 2 µm were stained with hematoxylin and eosin (H&E) to examine morphological changes. Staining was performed according to the procedure described elsewhere [32].

#### 4.5.2. Determination of DNA Damage (γH2AX)

Immunohistochemical analysis of the lung tissue sections from each animal (n = 36), each 2 μm thick, was performed to assess DNA damage. Antigen retrieval was conducted using citrate buffer (pH 6.0; Zytomed Systems GmbH, Berlin, Germany) in a steamer at 96 °C for 30 min. Prior to antibody application, non-specific binding was blocked with a rabbit Superblock solution (Zytomed Systems GmbH, Berlin, Germany) for 5 min. The sections were then incubated overnight at 4 °C with the Phospho-Histone H2AX (Ser139) (20E3) monoclonal antibody diluted 1:500 (Cell Signaling Technology, Frankfurt, Germany). After incubation, the slides were treated with the Histofine Simple Stain MAX PO (anti-rabbit) detection system for 30 min. After the washing steps, positive signals were visualized using AEC (Zytomed Systems GmbH, Berlin, Germany), and the tissues were counterstained with hematoxylin (Merck, Darmstadt, Germany). The intensity and distribution of staining were semi-quantitatively evaluated under a light microscope (All-in-One Microscope BZ-X800; Keyence, Neu-Isenburg, Germany). For quantification purposes, positive cells were counted in three representative fields of view per slide at 40× magnification across six animals per group. The data obtained from the PIC-treated groups (1B^1^–3B^1^) were compared to those from their respective PA control groups (1A^1^–3A^1^).

#### 4.5.3. Determination of Apoptosis (TUNEL)

The TUNEL assay, which detects apoptotic cells with extensive DNA fragmentation during the late stages of apoptosis, was performed using the DeadEnd™ Fluorometric TUNEL System (Promega GmbH, Walldorf, Germany). This assay was applied to 2 μm sections of lung samples from each mouse (n = 36), following the manufacturer’s instructions. The stained sections from the PIC-treated groups (1B^1^–3B^1^) were examined and evaluated using fluorescence microscopy (All-in-One Microscope BZ-X800; Keyence, Neu-Isenburg, Germany) and compared to those of their respective PA control groups (1A^1^–3A^1^).

#### 4.5.4. Determination of Lung Immune Cell Marker Alterations (CD45, Ly-6G, CD3, CD68, MCC)

Immunohistological staining was carried out on 2 μm sections of lung tissue to identify various immune cell populations within the tissue from each animal (n = 36). Specific primary antibodies targeting distinct immune cell markers were employed for this purpose. CD68 was used to detect monocytes and macrophages, while CD3 served as a marker for T lymphocytes (T-cells). To quantify all leukocytes, the marker CD45, also known as leukocyte common antigen, was used because of its expression on all leukocyte types. The marker Ly-6G was applied primarily to identify granulocytes, particularly neutrophils in mice. Additionally, an anti-mast cell chymase (MCC) antibody was used to detect mast cells present in lung tissue.

In all cases, antigen retrieval was carried using a citrate buffer (pH 6.0; Zytomed Systems GmbH, Berlin, Germany) at 96 °C for 30 min in a steamer. Before the application of the primary antibody, non-specific binding was minimized by incubating the samples with a blocking solution (rabbit Superblock; Zytomed Systems GmbH, Berlin, Germany) for 5 min. The primary antibodies—anti-CD68 (ab125212) at a 1:200 dilution (Abcam, Cambridge, UK), anti-CD3 (E4T1B) XP^®^ rabbit monoclonal antibody at 1:100 dilution (Cell Signaling Technology, Frankfurt, Germany), anti-CD45 (D3F8Q) rabbit monoclonal antibody at 1:100 dilution (Cell Signaling Technology, Frankfurt, Germany), anti-Ly-6G (E6Z1T) rabbit monoclonal antibody at 1:50 dilution (Cell Signaling Technology, Frankfurt, Germany), and anti-mast cell chymase (ab233103) at 1:20 dilution (Abcam, Cambridge, UK)—were then applied and incubated for 60 min at room temperature. Detection was performed using the Histofine Simple Stain MAX PO (anti-rabbit) universal immunoperoxidase polymer system (Medac Diagnostica, Wedel, Germany), following the manufacturer’s instructions. Positive staining was visualized with 3-Amino-9-Ethylcarbazole (AEC) (Zytomed Systems GmbH, Berlin, Germany). Finally, the tissue sections were counterstained with hematoxylin (Merck, Darmstadt, Germany), mounted with Aquatex (Merck, Darmstadt, Germany), and coverslipped. Quantification of the staining was performed as described in Section 4.5.2.

#### 4.5.5. Determination of Lung Fibrosis (NE)

Immunohistochemistry was performed using an anti-neutrophil elastase (NE) antibody to localize and quantify NE expression within lung tissue sections. This approach enables assessment of neutrophil infiltration and enzymatic activity in fibrotic regions. Lung tissue samples, sectioned into 2 μm slices from each animal (n = 36), were first pretreated by steaming for 30 min at 96 °C in citrate buffer (pH 6.0; Zytomed Systems GmbH, Berlin, Germany). To prevent non-specific binding, the sections were incubated with a blocking solution containing rabbit Superblock (Zytomed Systems GmbH, Berlin, Germany) for 5 min. Next, the sections were exposed to a primary antibody—anti-NE rabbit monoclonal antibody (E8U3X) diluted 1:200 (Cell Signaling Technology, Frankfurt, Germany)—and incubated for one hour at room temperature. The samples were then treated with Histofine Simple Stain MAX PO (anti-rabbit) for 30 min to detect the bound primary antibody. After thorough washing, positive signals were visualized using AEC substrate (Zytomed Systems GmbH, Berlin, Germany), and the tissue was counterstained with hematoxylin (Merck, Darmstadt, Germany). Quantification of the staining was performed as described in Section 4.5.2.

### 4.6. Isolation of Ribonucleic Acid (RNA) and Reverse Transcription

RNA was isolated from 250,000 PA- (group 1A–3A) or PIC- (group 1B–3B) treated BALF cells or from lung tissue (group 1A^1^–3A^1^ and group 1B^1^–3B^1^) using the NucleoSpin^®^ RNA Plus Kit (Macherey-Nagel, Düren, Germany) according to the manufacturer’s instructions. cDNA was generated with the SuperScript II Reverse Transcriptase kit (Invitrogen; Thermo Fisher Scientific) using 2–5 µg of total RNA for transcription according to the manufacturer’s instructions.

### 4.7. Quantitative Real-Time Polymerase Chain Reaction (PCR) Analysis

The gene expression analysis was conducted using quantitative real-time PCR with specific primer sets obtained from Sigma-Aldrich (Steinheim, Germany) and under the conditions outlined in Table 2. The reactions were performed using LightCycler technology (Roche Diagnostics, Mannheim, Germany), following the protocol described previously [33]. Melting curve analysis was employed to assess the specificity of the PCR products. To verify cDNA quality and to serve as a normalization control, beta-actin (β-actin) was amplified in all samples. Each sample was tested in duplicate to ensure accuracy.

### 4.8. Statistical Analysis

All data were analyzed with GraphPad Prism software (GraphPad Software Inc., San Diego, CA, USA) version 10.4.1. and expressed as mean ± standard deviation (SD). Ordinary one-way ANOVA with Tukey’s multiple comparison test was done to indicate differences in the mean within the groups (1A–3A and 1B–3B or 1A^1^–3A^1^ and 1B^1^–3B^1^). An unpaired *t* test was used to compare the PA control group within the corresponding PIC treated group (1A/1B, 2A/2B, 3A/3B and 1A^1^/1B^1^, 2A^1^/2B^1^, 3A^1^/3B^1^). Significant results are indicated * *p* < 0.05, ** *p* < 0.01, *** *p* < 0.001, or **** *p* < 0.0001; ns (not significant).

## 5. Conclusions

In conclusion, plasma intensive care (PIC) treatment induces transient immune responses in the lungs, such as increased cellular activation, apoptosis, and cytokine mRNA expression. These responses appear to be reversible within a certain recovery period, with cellular composition and viability returning to baseline levels. These findings indicate that PIC is likely to be safe for respiratory applications when appropriately monitored. The observed elevation in IFN-γ suggests potential immunomodulatory benefits, possibly enhancing antimicrobial defenses. However, the induction of inflammatory markers underscores the importance of careful regulation and ongoing assessment to mitigate risks associated with excessive inflammation. Overall, these findings support the continued investigation of PIC as a promising therapeutic approach with a manageable safety profile.

## 6. Patents

WO2022008684A1 (System and plasma for treating and/or preventing a viral, bacterial and/or fungal infection).

## Figures and Tables

**Figure 1 ijms-26-08852-f001:**
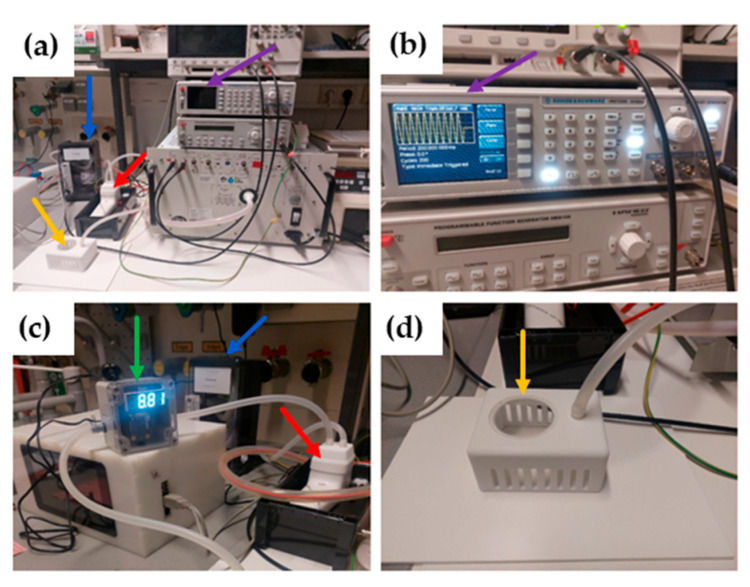
Plasma intensive care (PIC) setup. (**a**) The complete treatment assembly shows the CAP device (red arrow), the treatment chamber (yellow arrow), the function generator (purple arrow), and the pressurized air generator with an integrated flowmeter (blue arrow). (**b**) Displays the function generator, which allows for the modification of the plasma parameters. (**c**) A close-up image of the CAP device (red arrow) with the pressurized air generator in the background (blue arrow) and the integrated ozone sensor (green arrow). (**d**) The treatment chamber features an integrated plastic window for visual monitoring of the animals during the experiment, as well as ventilation slits. A hose connected to the ozone sensor leads into the treatment chamber, directing CAP components from the plasma source to the treatment chamber.

**Figure 2 ijms-26-08852-f002:**
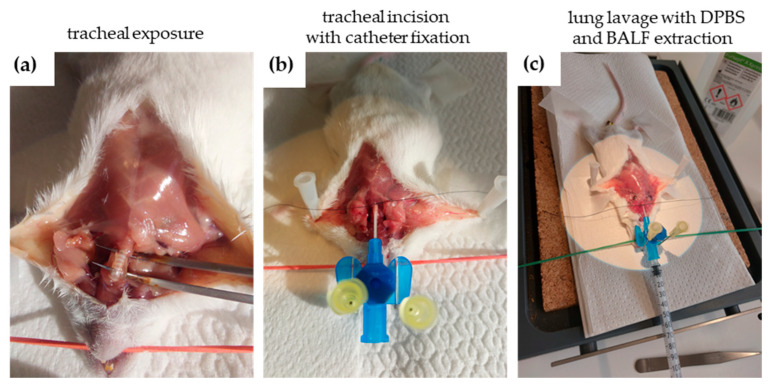
Procedure for collecting bronchoalveolar lavage fluid (BALF) from an animal post-mortem. (**a**) After placing the animal in a dorsal position and disinfecting the area, a surgical incision is made above the thymus to expose the trachea. (**b**) The trachea is then cannulated with a tube secured with sutures. Using a microscope, an incision is made between tracheal rings, and the tube is inserted into the airways. (**c**) BALF is collected by injecting Dulbecco’s Phosphate-Buffered Saline (DPBS) into the trachea and by gently massaging the chest to facilitate fluid retrieval, repeating this process three times to obtain about 2 mL of fluid. The collected BALF is centrifuged to separate immune cells from fluids for further analysis.

**Figure 3 ijms-26-08852-f003:**
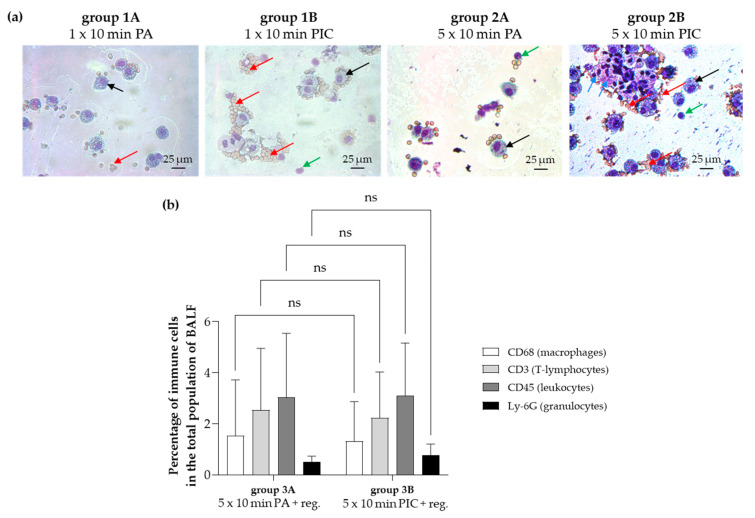
Immune cell examination from bronchoalveolar lavage fluid (BALF) extraction after PA and PIC treatment. (**a**) Cytocentrifuge preparation of cells obtained by lavage from short-term PA (group 1A) and PIC (group 1B) treatment and from long-term PA (group 2A) and PIC (group 2B) treatment. The micrographs demonstrate the presence of macrophages (black arrows), lymphocytes (green arrows), and erythrocytes (red arrows). After PIC treatment, cell cluster formation and erythrocyte adhesion was increased. (**b**) Percentage of immune cells (macrophages, T-lymphocytes, leukocytes, and granulocytes) in the total BALF population after long-term recovery-phase treatments (group 3A and 3B). Statistical analysis: Unpaired *t* test was done to compare the mean of group 3A to group 3B. ns (not significant).

**Figure 4 ijms-26-08852-f004:**
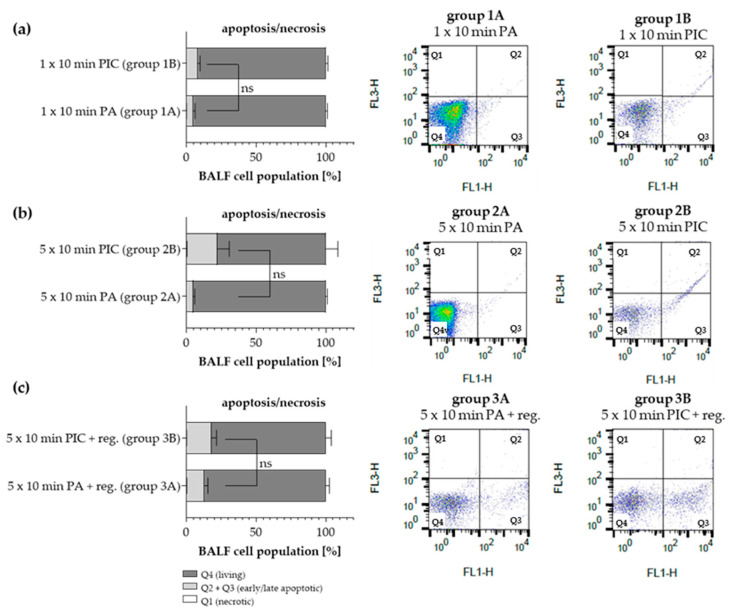
Annexin V/PI double-staining assay was performed after PA and PIC treatment. Apoptosis (early and late apoptosis), necrosis, and the number of live cells from BALF extraction were analyzed after (**a**) short-term PA (group 1A) and PIC (group 1B) treatment (**b**) long-term PA (group 2A) and PIC (group 2B) treatment, and (**c**) long-term PA (group 3A) and PIC (group 3B) treatment with a regeneration phase of 7 days. Statistical analysis: Unpaired *t* test was done to compare the mean of group A to group B. ns (not significant). The graphs present the percentage (mean +/− SD) of the cells in the region among the total cells from n = 6 mice per group. The corresponding plots show a typical result from a representative mouse per group. The y-axis (FL3-H) shows the PI-labeled population and the x-axis (FL1-H) the FITC-labeled Annexin V positive cells. Necrotic cells are Annexin V negative and PI positive (top-left sector (Q1)), apoptotic cells are both PI and Annexin V positive (top-right sector (Q2)), pre-apoptotic cells are Annexin V positive and PI negative (lower-right sector (Q3)), whereas live cells are PI and Annexin V negative (lower-left sector (Q4)).

**Figure 5 ijms-26-08852-f005:**
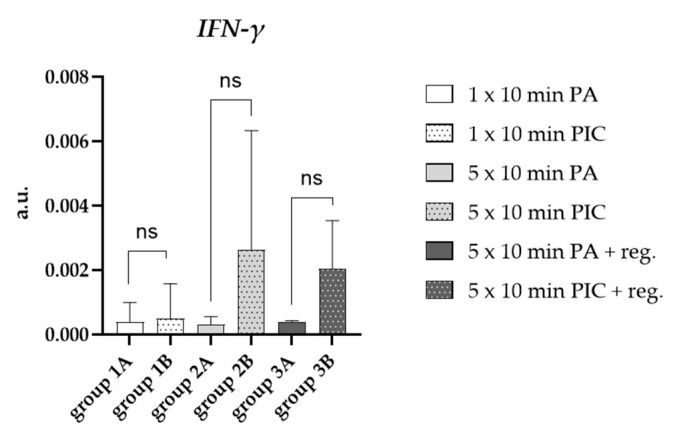
Effects of PIC treatment on pro-inflammatory gene expression in BALF cells. The relative mRNA expression in arbitrary units (a.u.) of IFN-γ was analyzed after short-term PA (group 1A) and PIC (group 1B) treatment, after long-term PA (group 2A) and PIC (group 2B) treatment, and after long-term PA (group 3A) and PIC (group 3B) treatment with a regeneration phase (reg.) of 7 days. Statistical analysis: Unpaired *t* test was done to compare the mean of group A to group B. ns (not significant).

**Figure 6 ijms-26-08852-f006:**
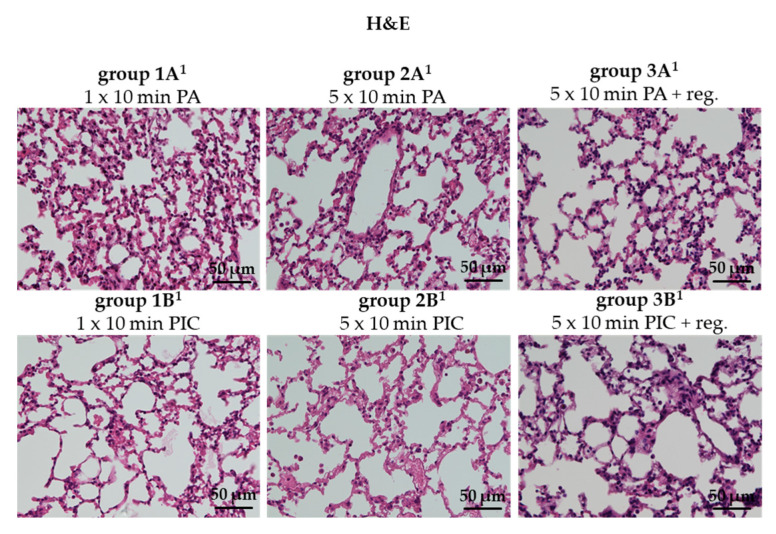
Hematoxylin and eosin (H&E) staining of lung tissue sections after short-term PA (group 1A^1^) and PIC (group 1B^1^) treatment, after long-term PA (group 2A^1^) and PIC (group 2B^1^) treatment, and after long-term PA (group 3A^1^) and PIC (group 3B^1^) treatment with a regeneration phase (reg.) of 7 days. Representative images of each group are presented at 40× magnification.

**Figure 7 ijms-26-08852-f007:**
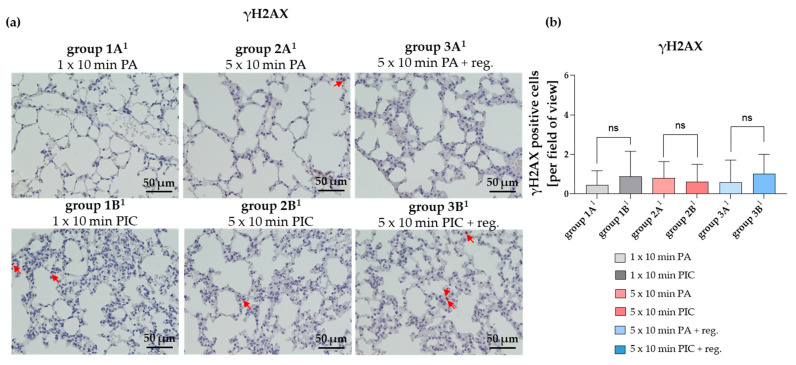
Gamma H2AX (γH2AX) histological staining of lung tissue sections after short-term PA (group 1A^1^) and PIC (group 1B^1^) treatment, after long-term PA (group 2A^1^) and PIC (group 2B^1^) treatment, and after long-term PA (group 3A^1^) and PIC (group 3B^1^) treatment with a regeneration phase (reg.) of 7 days. (**a**) Representative images of each group are presented at 40× magnification. Red arrows indicate γH2AX positive cells. (**b**) The positive cells from three fields of view per animal were summarized and analyzed per treatment group. Statistical analysis: Unpaired *t* test was done to compare the mean of group A^1^ to the corresponding group B^1^. ns (not significant).

**Figure 8 ijms-26-08852-f008:**
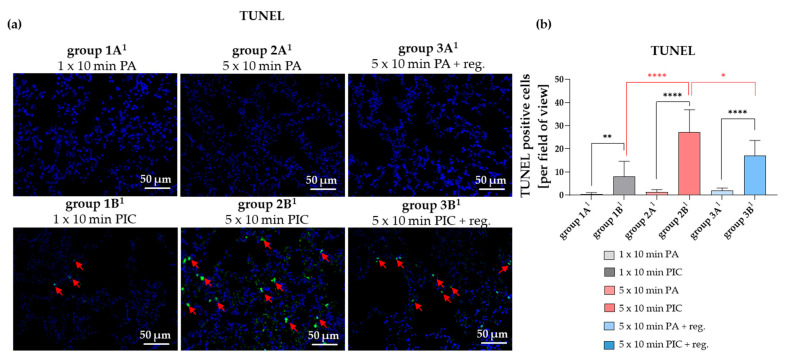
TUNEL staining of lung tissue sections after short-term PA (group 1A^1^) and PIC (group 1B^1^) treatment, after long-term PA (group 2A^1^) and PIC (group 2B^1^) treatment, and after long-term PA (group 3A^1^) and PIC (group 3B^1^) treatment with a regeneration phase (reg.) of 7 days. (**a**) Representative images of each group are presented at 40× magnification. Red arrows indicate apoptotic cells. Apoptotic cells are green, and DAPI staining is blue. (**b**) The positive cells from three fields of view per animal were summarized and analyzed per treatment group. Statistical analysis: Unpaired *t* test was done to compare the mean of group A^1^ to the corresponding group B^1^. One way ANOVA with Tukey’s multiple comparison test was done to compare the mean of group 1B^1^ to 2B^1^ and of group 2B^1^ to 3B^1^. * *p* ≤ 0.05, ** *p* < 0.01, **** *p* < 0.0001.

**Figure 9 ijms-26-08852-f009:**
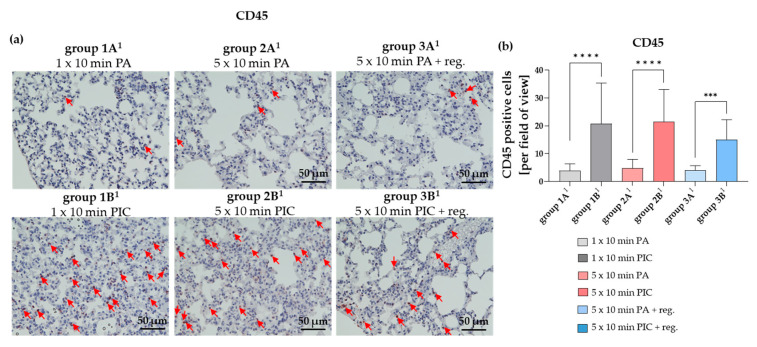
CD45 histological staining of lung tissue sections after short-term PA (group 1A^1^) and PIC (group 1B^1^) treatment, after long-term PA (group 2A^1^) and PIC (group 2B^1^) treatment, and after long-term PA (group 3A^1^) and PIC (group 3B^1^) treatment with a regeneration phase (reg.) of 7 days. (**a**) Representative images of each group are presented at 40× magnification. Red arrows indicate CD45 positive cells. (**b**) The positive cells from three fields of view per animal were summarized and analyzed per treatment group. Statistical analysis: Unpaired *t* test was done to compare the mean of group A^1^ to the corresponding group B^1^. *** *p* < 0.001, **** *p* < 0.0001.

**Figure 10 ijms-26-08852-f010:**
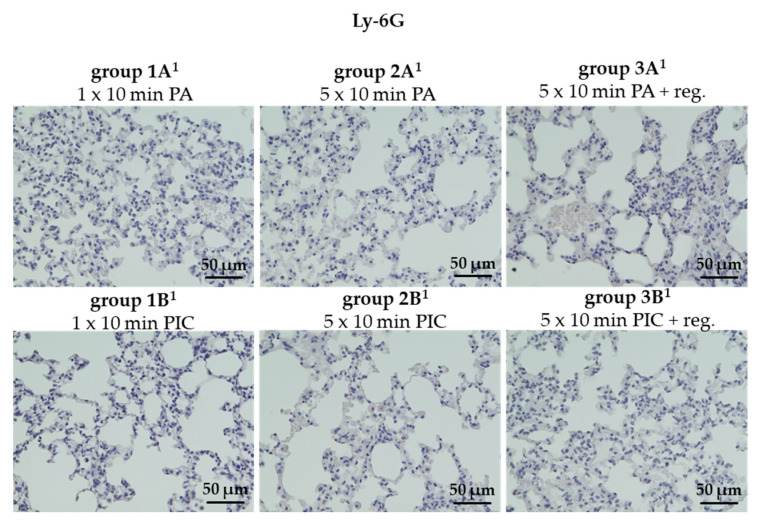
Ly-6G histological staining of lung tissue sections after short-term PA (group 1A^1^) and PIC (group 1B^1^) treatment, after long-term PA (group 2A^1^) and PIC (group 2B^1^) treatment, and after long-term PA (group 3A^1^) and PIC (group 3B^1^) treatment with a regeneration phase (reg.) of 7 days. Representative images of each group are presented at 40× magnification. Positive cells were not detectable in any of the sections.

**Figure 11 ijms-26-08852-f011:**
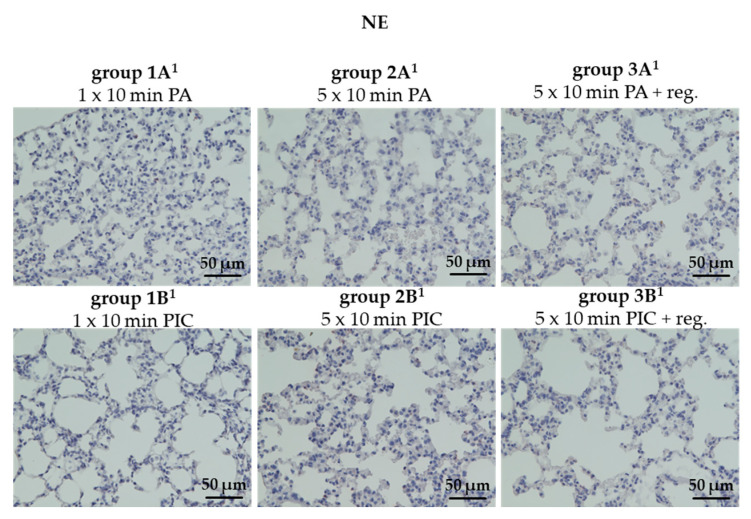
Neutrophil elastase (NE) histological staining of lung tissue sections after short-term PA (group 1A^1^) and PIC (group 1B^1^) treatment, after long-term PA (group 2A^1^) and PIC (group 2B^1^) treatment, and after long-term PA (group 3A^1^) and PIC (group 3B^1^) treatment with a regeneration phase (reg.) of 7 days. Representative images of each group are presented at 40× magnification. Positive stainings were not detectable.

**Figure 12 ijms-26-08852-f012:**
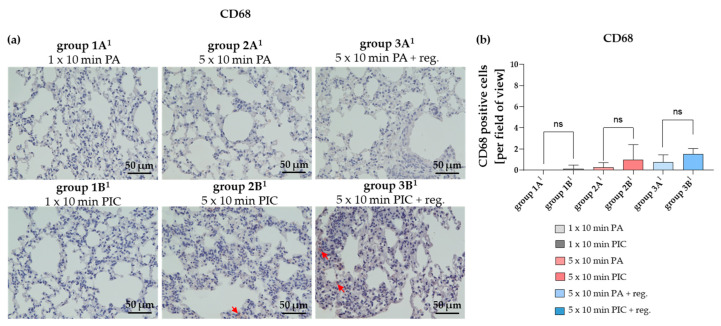
CD68 histological staining of lung tissue sections after short-term PA (group 1A^1^) and PIC (group 1B^1^) treatment, after long-term PA (group 2A^1^) and PIC (group 2B^1^) treatment, and after long-term PA (group 3A^1^) and PIC (group 3B^1^) treatment with a regeneration phase (reg.) of 7 days. (**a**) Representative images of each group are presented at 40× magnification. Red arrows indicate CD68 positive cells. (**b**) The positive cells from three fields of view per animal were summarized and analyzed per treatment group. Statistical analysis: Unpaired *t* test was done to compare the mean of group A^1^ to the corresponding group B^1^. (ns) not significant.

**Figure 13 ijms-26-08852-f013:**
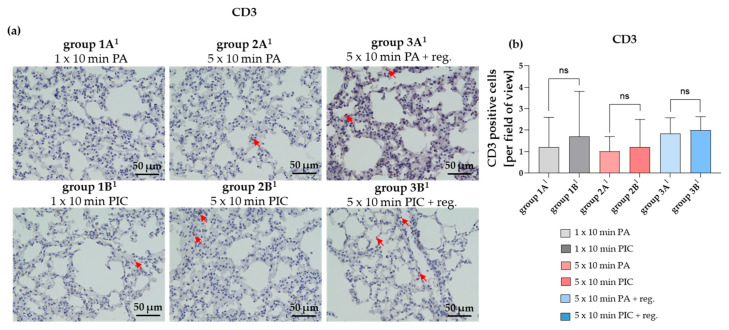
CD3 histological staining of lung tissue sections after short-term PA (group 1A^1^) and PIC (group 1B^1^) treatment, after long-term PA (group 2A^1^) and PIC (group 2B^1^) treatment, and after long-term PA (group 3A^1^) and PIC (group 3B^1^) treatment with a regeneration phase (reg.) of 7 days. (**a**) Representative images of each group are presented at 40× magnification. Red arrows indicate CD3 positive cells. (**b**) The positive cells from three fields of view per animal were summarized and analyzed per treatment group. Statistical analysis: Unpaired *t* test was done to compare the mean of group A^1^ to the corresponding group B^1^. (ns) not significant.

**Figure 14 ijms-26-08852-f014:**
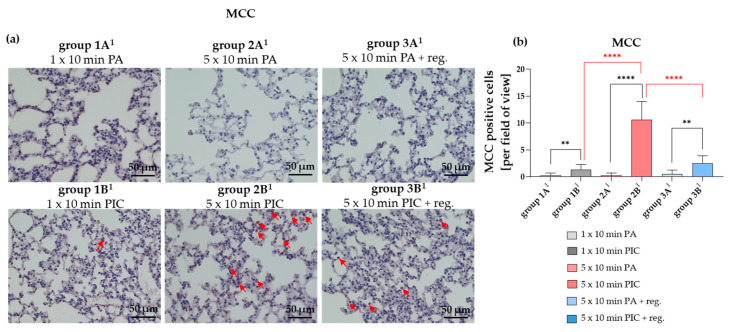
Mast cell chymase (MCC) histological staining of lung tissue sections after short-term PA (group 1A^1^) and PIC (group 1B^1^) treatment, after long-term PA (group 2A^1^) and PIC (group 2B^1^) treatment, and after long-term PA (group 3A^1^) and PIC (group 3B^1^) treatment with a regeneration phase (reg.) of 7 days. (**a**) Representative images of each group are presented at 40× magnification. Red arrows indicate MCC positive cells. (**b**) The positive cells from three fields of view per animal were summarized and analyzed per treatment group. Statistical analysis: Unpaired *t* test was done to compare the mean of group A^1^ to the corresponding group B^1^. One way ANOVA with Tukey’s multiple comparison test was done to compare the mean of group 1B^1^ to 2B^1^ and of group 2B^1^ to 3B^1^. ** *p* < 0.01, **** *p* < 0.0001.

**Figure 15 ijms-26-08852-f015:**
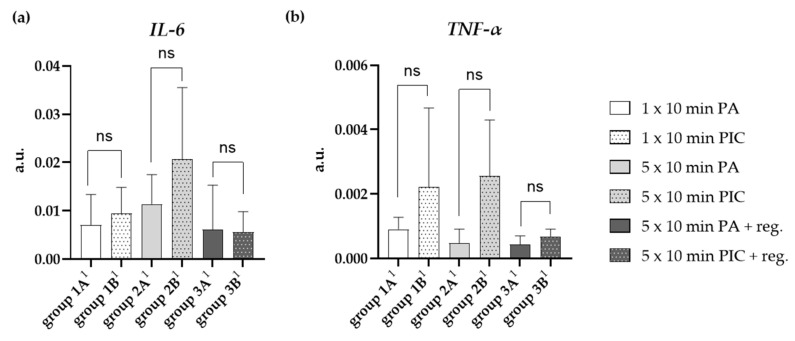
Effects of PIC treatment on pro-inflammatory gene expression in lung tissue. The relative mRNA expression in arbitrary units (a.u.) of (**a**) *IL-6* and (**b**) *TNF-α* was analyzed after short-term PA (group 1A^1^) and PIC (group 1B^1^) treatment, after long-term PA (group 2A^1^) and PIC (group 2B^1^) treatment, and after long-term PA (group 3A^1^) and PIC (group 3B^1^) treatment with a regeneration phase (reg.) of 7 days. Statistical analysis: Unpaired *t* test was done to compare the mean of group A^1^ to group B^1^. ns (not significant).

**Figure 16 ijms-26-08852-f016:**
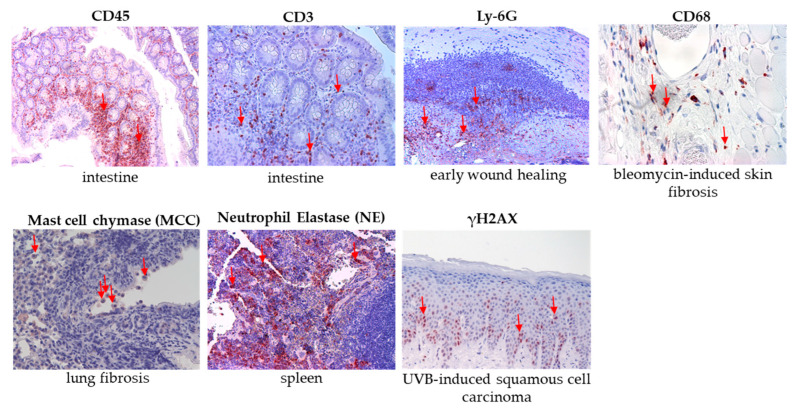
Immunohistochemistry positive controls for the antibodies used in the study. Intestinal tissue was used for CD45 and CD3 staining. Tissue from early wound healing was used for Ly-6G staining. Bleomycin-induced skin fibrosis sections were used for CD68 detection. Fibrotic lung tissue sections were used for mast cell chymase staining, spleen tissue for neutrophil elastase staining and UVB-induced squamous cell carcinoma sections were used for γH2AX detection. Red arrows indicate positive cells. Representative images are presented at 20× magnification (CD45, CD3, Ly-6G, MCC, NE, γH2AX) and at 40× magnification (CD68).

**Table 1 ijms-26-08852-t001:** Animal groups with treatment regime and experimental evaluation.

Group (n = 6)	Treatment	Evaluation
1A	1 × 10 min PA	BALF
1B	1 × 10 min PIC	BALF
2A	5 × 10 min PA	BALF
2B	5 × 10 min PIC	BALF
3A	5 × 10 min PA; 7 days reg.	BALF
3B	5 × 10 min PIC; 7 days reg.	BALF
1A^1^	1 × 10 min PA	histology
1B^1^	1 × 10 min PIC	histology
2A^1^	5 × 10 min PA	histology
2B^1^	5 × 10 min PIC	histology
3A^1^	5 × 10 min PA; 7 days reg.	histology
3B^1^	5 × 10 min PIC; 7 days reg.	histology

^1^ Separate groups with the same treatment but with a different post hoc evaluation. BALF: bronchoalveolar lavage fluid; PA: pressurized air; PIC: plasma intensive care.

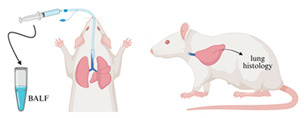

**Table 2 ijms-26-08852-t002:** Murine primers and conditions.

Primer Name	Forward Primer 5′ → 3′	Reverse Primer 5′ → 3	Condition ^1^
β-actin	AGTGTGACGTTGACATCCGT	GTAACAGTCCGCCTAGAAGC	ann. 60 °C
			melt. 81 °C
IL-6	GTCCTTCCTACCCCAATTTCCA	TAACGCACTAGGTTTGCCGA	ann. 60 °C
			melt. 77 °C
TNF-α	AGCCCACGTCGTAGCAAACC	CGGGGCAGCCTTGTCCCTTG	ann. 60 °C
			melt. 84 °C
IFN-γ	AGCAAGGCGAAAAAGGATGC	CTCATTGAATGCTTGGCGCT	ann. 60 °C
			melt. 77 °C

^1^ ann.: annealing; melt.: melting.

## Data Availability

Data are available upon reasonable request from the corresponding author.

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
