# Peer review of "In Vivo Immune Cell Responses and Long-Term Effects of Cold Atmospheric Plasma in the Upper Respiratory Tract"

_ijms, 2025, doi:10.3390/ijms26188852_

Round 1

Reviewer 1 Report

Comments and Suggestions for Authors

The submitted paper evaluates the effects of Cold Atmospheric Plasma (CAP) from a Plasma Intensive Care (PIC) device on the Upper Respiratory Tract (URT), with a focus on immune cells responses and long term effects. The study is performed in vivo on a population of 72 mice in groups of 6.

The topic is particularly relevant, given the promising therapeutic and preventive infection control capabilities of CAP and its applicability to difficult to access anatomical structures like the URT. This work contributes to investigate the potential side effects, particularly excessive immune responses or cytotoxicity. The results of the work are promising, as only transient immune responses are produced in the lungs, which appear to be reversible, thus indicating PIC as a safe therapy to be carefully regulated in order to mitigate risks associated with excessive inflammation.

The paper is clearly written and well structured, properly set into the context and supported by suitable references.

Only minor comments are proposed mainly for clarification:

Title – Though already quite long, an effort could be made to include the term “in vivo” in the title, as it is an essential characteristic of this work

Line 84 “Their findings indicate that treatment with the PIC device has a pronounced impact on the functionality and activity of isolated PMNs” – can you please be more specific? What kind of “pronounced impact”? positive or negative?

Apart for this specific comment, few lines could be added at the end of this Introduction to more generally comment the balance of positive and potentially negative effects of immune responses stimulation by CAP.

Figure 7 – what do the red arrows indicate? It is not explained in the caption or main text

Line 489 “n = 6 mice per group” – can you please add a short comment on the statistical significance of this choice (n=6), considering also that many unpaired t tests resulted not significant?

Line 502 “with a 4 kHz setting specifically used in this study” – why the oxygen mode (4 kHz) was preferred in this study ?

Line 520 “period (200 ms), and number of cycles (140)” – please shortly explain the impact of these settings and how they link to the 10 minutes duration of a single session

Line 532 “once a week for 10 minutes” – for how many weeks? What is the relevance of the total duration?

Reviewer 2 Report

Comments and Suggestions for Authors

The manuscript is well written and addresses a novel and relevant topic. Nevertheless, the following points should be carefully considered:

  1. The apoptosis results showed only a trend, without statistical significance. Moreover, the plot presented represents only one animal per group. This information should be explicitly stated in the Results and further addressed in the Discussion.
  2. Inflammatory cells are observed in all groups, with higher numbers in groups 3A and 3B. If this pattern was not consistently reproduced across the remaining histological sections, the image provided may not be representative. In this case, the Results and Discussion should better clarify that, although some degree of inflammation was present, it did not differ from the respective controls. A more descriptive histological analysis is also expected, specifying which inflammatory cells were identified within the parenchyma and the degree of inflammation (mild, moderate, or severe) in each group.
  3. Please clarify whether positive controls were included in the immunohistochemistry assays to ensure that the reactions were properly validated. This information should be explicitly reported in the Results, particularly in cases where staining was absent or minimal, and the Methods should specify which positive controls were employed.
  4. In the mast cell graph, why was the comparison between groups 2B and 3B omitted, whereas such a comparison was performed in the TUNEL analysis? The text states that the increase after the regeneration period was discrete, but as a statistically significant difference was detected compared to the control, this finding should be considered
  5. The results and discussion sections should state that gene expression was evaluated, rather than cytokine protein levels (IL-6 and TNF-α). It is also important to emphasize that only a trend toward increased expression was observed, without statistical significance.
  6. In lines 431–433 of the Discussion, the authors state that the same CD-45 results were not observed in BALF, suggesting that the treatment does not affect bronchial tissue. However, BALF is a fluid, not a tissue. Its representativeness of tissue response should be discussed.
  7. At the end of the Discussion, the authors conclude: “PIC treatment appears to elicit a temporary inflammatory response that resolves over time without causing any lasting tissue damage, supporting its potential safety profile for therapeutic applications.” However, although inflammatory signs diminished, they were still detectable after the recovery period. This information should be incorporated into the final interpretation.

Round 2

Reviewer 2 Report

Comments and Suggestions for Authors

“Comments 2: Inflammatory cells are observed in all groups, with higher numbers in groups 3A and 3B. If this pattern was not consistently reproduced across the remaining histological sections, the image provided may not be representative. In this case, the Results and Discussion should better clarify that, although some degree of inflammation was present, it did not differ from the respective controls. A more descriptive histological analysis is also expected, specifying which inflammatory cells were identified within the parenchyma and the degree of inflammation (mild, moderate, or severe) in each group.

Response 2: Unfortunately, there is no direct reference to a specific figure here. We suspect that the reviewer may be referring to CD3 histology (Figure 13), as inflammatory cells were detected across all groups, with a particularly higher number of positive cells observed in groups 3A and 3B. We agree with the reviewer that the selection of histological images does not always precisely correspond to the corresponding overall quantification shown in the figure. Especially when only a very small number of positive cells are detectable, selecting images that are representative of the entire set of histological sections for a treatment group or the respective control group is particularly challenging. The histological images we selected may suggest that they are not representative of the overall tissue response. We have explicitly described this issue in the Results section and again highlighted it in the Discussion. We hope that our additions have satisfied the reviewer's requirements.

“However, it should be noted here that the overall low numbers of positive cells in the selected images may not fully represent all slices for a treatment or control group. Despite this, there is a trend toward a slight increase in CD3 after PIC treatment. PIC treatment (groups 1B1, 2B1, 3B1) likely induces at most a mild inflammatory response when considering the full dataset. The higher apparent inflammation in control group 3A1 is due to greater sample-to-sample variability rather than a consistently higher CD3 count. However, quantitative analysis across all sections showed no statistically significant difference in inflammatory cell density, suggesting that T lymphocyte populations are not significantly affected by PIC treatment.” (line 336-344)

“Even though the results of the CD3 assessment suggest a mild inflammatory response after PIC treatment (Figure 13), it should be noted that these results indicate only a trend due to the high variability among individual samples and do not provide statistically significant findings. Therefore, these results should be interpreted with caution.” (line 490-493) “

RESPONSE: I am referring to figure 6 (HE staining). Please, check it.
